# Persistent biotic interactions of a Gondwanan conifer from Cretaceous Patagonia to modern Malesia

Michael P. Donovan [1,2,3✉], Peter Wilf [3], Ari Iglesias [4], N. Rubén Cúneo[5] & Conrad C. Labandeira[2,6,7]

Many plant genera in the tropical West Pacific are survivors from the paleo-rainforests of Gondwana. For example, the oldest fossils of the Malesian and Australasian conifer *Agathis* (Araucariaceae) come from the early Paleocene and possibly latest Cretaceous of Patagonia, Argentina (West Gondwana). However, it is unknown whether dependent ecological guilds or lineages of associated insects and fungi persisted on Gondwanan host plants like *Agathis* through time and space. We report insect-feeding and fungal damage on Patagonian *Agathis* fossils from four latest Cretaceous to middle Eocene floras spanning ca. 18 Myr and compare it with damage on extant *Agathis*. Very similar damage was found on fossil and modern *Agathis*, including blotch mines representing the first known Cretaceous–Paleogene boundary crossing leaf-mine association, external foliage feeding, galls, possible armored scale insect (Diaspididae) covers, and a rust fungus (Pucciniales). The similar suite of damage, unique to fossil and extant *Agathis*, suggests persistence of ecological guilds and possibly the component communities associated with *Agathis* since the late Mesozoic, implying host tracking of the genus across major plate movements that led to survival at great distances. The living associations, mostly made by still-unknown culprits, point to previously unrecognized biodiversity and evolutionary history in threatened rainforest ecosystems.

[1] Department of Paleobotany and Paleoecology, Cleveland Museum of Natural History, 1 Wade Oval Drive, Cleveland, OH 44106, USA. [2] Department of Paleobiology, National Museum of Natural History, Smithsonian Institution, P.O. Box 37012, Washington, DC 20013, USA. [3] Department of Geosciences and Earth and Environmental Systems Institute, Pennsylvania State University, 537 Deike Building, University Park, PA 16802, USA. [4] Instituto de Investigaciones en Biodiversidad y Medioambiente, CONICET-Universidad Nacional del Comahue, Quintral 1250, San Carlos de Bariloche, 8400 Río Negro, Argentina. [5] CONICET-Museo Paleontológico Egidio Feruglio, Av. Fontana 140, 9100 Trelew, Chubut, Argentina. [6] Department of Entomology and Behavior, Ecology, Evolution, and Systematics Program, University of Maryland, College Park, MD 20742, USA. [7] College of Life Sciences, Capital Normal University, 100048 Beijing, China. ✉email: mdonovan@cmnh.org

Terminal Cretaceous to Eocene floras from Patagonia, Argentina show affinities with fossil floras from Australia and other southern land masses, as well as living subtropical and tropical montane rainforest floras from Australasia to Southeast Asia[1–8]. These biogeographic connections derive from the former presence of mesic austral rainforests toward the end of Gondwana, which were nearly exterminated following the final separation of South America, Antarctica, and Australia starting from the middle Eocene. Many of the survivor lineages were sheltered in Australia as the continent moved northward to its eventual, late Oligocene collision with Asia, initiating new biotic interchanges[4,5,9–11]. A number of angiosperm[12–19], fern[20], and conifer[21–23] genera with latest Cretaceous to Eocene fossil records in Patagonia exhibit this pattern of Old World survival[5]. The early Paleogene floras of Gondwanan Patagonia hosted diverse insect-herbivore communities based on the richness of insect feeding damage on fossil leaves[24–26] and body fossils[27–29], but whether the ancient insect communities or ecological associations tracked their host plants across later range shifts has been unknown.

Recently, the first South American members of the Old World broadleaved conifer genus *Agathis* (kauris, dammars; Araucariaceae), including its oldest known representatives, were recognized from multiple vegetative and reproductive organs found in early Paleocene[7], early and middle Eocene[22], and probably in terminal Cretaceous[7] floras of central Patagonia, Argentina. The fossils document the evolution of the genus from an early Paleocene stem lineage to an Eocene member of the *Agathis* crown[7]. Older fossils of the agathioid clade belong to extinct taxa that lack distinctive characters of its two living genera, *Wollemia* and *Agathis*[7,30–32]. All previous *Agathis* fossils had come from the late Paleocene to early Miocene of Australia and the late Oligocene–Miocene of New Zealand, with no reports of unequivocal associated insect damage[33–35]. The separation of South America and Antarctica and associated cooling and aridification[36,37] presumably led to the loss of suitable habitat and the extinction of *Agathis* in South America along with other paleo-Antarctic genera[5], but the genus persists today with ca. 17 species in lowland to upper montane rainforests in Australasia and across Wallace's Line to Sumatra in Malesia[38]. *Agathis* trees are large-statured keystone species[38], often emerging prominently above angiosperms in the canopy, and they host diverse animal and fungal communities[39,40]. The genus faces immense logging pressure through much of its modern range[38].

The Patagonian *Agathis* material includes abundant, well-preserved leaf-compression fossils[7,22], which we observed to be associated with diverse insect herbivore damage types (DTs), including distinctive blotch mines (Fig. 1a–i and Supplementary Figs. 1–4), that resemble those we have seen on extant *Agathis* species (Fig. 1j–m and Supplementary Fig. 5). Although some extant *Agathis* insect associations have been studied (Supplementary Data 1), most extant associations that resemble the fossils are undocumented, indicating undiscovered biodiversity and evolutionary history that may exist as ancient herbivore and fungal associations surviving on modern *Agathis* hosts. The high relative abundances of the fossils studied here[7,11,22] indicate that *Agathis* was a dominant genus in the past like it is today[41]; thus, the fossils represent ecologically important component communities that were present during the early evolution of the genus. To address the possibilities of the persistence of insect herbivore and fungal ecological guilds and the possible tracking of *Agathis* through time and space by component communities, we compare insect and fungal damage on fossil *Agathis* from Patagonia with damage on extant herbarium collections and living, wild *Agathis* trees.

We examined leafy branches, isolated leaves, and male and female reproductive organs of the well-preserved Argentine fossils *Agathis immortalis*[7], a stem-group *Agathis* from the early Paleocene Palacio de los Loros 2 locality (PL2; Salamanca Formation; geomagnetic polarity chron C28n, which has an age range of 64.47–63.49 Ma)[7,42–44]; and *Agathis zamunerae*[22], a crown-group *Agathis*[7] from the early Eocene Laguna del Hunco (LH; Huitrera Formation; ca. 52.2 Ma) and middle Eocene Río Pichileufú (RP; Huitrera Formation; ca. 47.7 Ma) localities[11,21]. In addition, we examined cf. *Agathis* sp. leaves[7] from the latest Cretaceous (Maastrichtian; 67–66 Ma) portion of the Lefipán Formation[45,46]. The Cretaceous specimens, only known from isolated leaves, exhibit morphological characters consistent with *Agathis*, although no cuticle or reproductive organs have been recovered[7]. All sites are located in Patagonian Argentina, in Chubut (Lefipán, PL2, LH) and Río Negro (RP) provinces. Together, these localities represent four time slices spanning ca. 18 million years of the early evolution of *Agathis* and its component communities, which are defined as assemblages of ecologically dependent component species occurring on a particular plant-host species[47]. We found a suite of diverse biotic associations never before seen on fossil *Agathis*, and we erect a new leaf mining ichnogenus and two ichnospecies; discuss other damage, including external foliage feeding, galling, possible armored scale insect (Hemiptera: Diaspididae) covers, and epiphyllous fungi (Pucciniales); and compare the damage from Patagonia with its modern analogs on extant *Agathis*. From the survey of modern material, we also report a rich, largely undocumented array of modern insect herbivory on living *Agathis* across its range that is similar to the new fossil exemplars.

## Results
### Systematic paleontology

*Frondicuniculum* ichnogen. nov.

**Etymology**. Classical Latin: *frons -dis*, a leaf, leafy twig or foliage; and *cuniculum-i*, meaning a mine, underground passage, hole or pit.
**Type ichnospecies**. *Frondicuniculum lineacurvum* ichnosp. nov.
**Diagnosis**. Elongate-ellipsoidal blotch mines occurring on broadleaved, parallel veined conifer leaves. Long axes of the mines are parallel to leaf venation. Frass, when present, is densely packed, composed of spheroidal pellets surrounded by amorphous matter, often positioned along one margin of the mine. Leaf veins within mines are distorted.

*Frondicuniculum lineacurvum* ichnosp. nov.

**Etymology**. Classical Latin: *linea-e*, a string, linen thread, or drawn line; and *curvus–a –um*, bent, bowed, arched, or curved.
**Holotype**. MPEF-Pb 6336 (Fig. 1d–f and Supplementary Fig. 3a–e), Laguna del Hunco quarry LH6[10], early Eocene, Chubut Province, Argentina.
**Paratypes**. MPEF-Pb 3160 (Laguna del Hunco quarry LH6, Supplementary Fig. 3f), USNM 545226 (Río Pichileufú historical collection[48], Fig. 1g, h and Supplementary Fig. 4a, b).
**Diagnosis**. As for the genus, with smooth, linear to gently curving mine margins.
**Description**. An elongate-ellipsoidal blotch mine positioned along leaf margin, long axis parallel to leaf veins, mine margins well defined, linear to gently curving. Mine dimensions 7.2–50.0 mm long by 2.0–10.0 mm wide. Frass, when present, composed of spherical or hemispherical pellets measuring 0.04–0.12 mm in diameter and surrounded by dark, amorphous matter. Frass pellets mostly positioned near mine margin and may be replaced by surrounding amber with original frass material not preserved. Reaction rim 0.1–0.3 mm wide

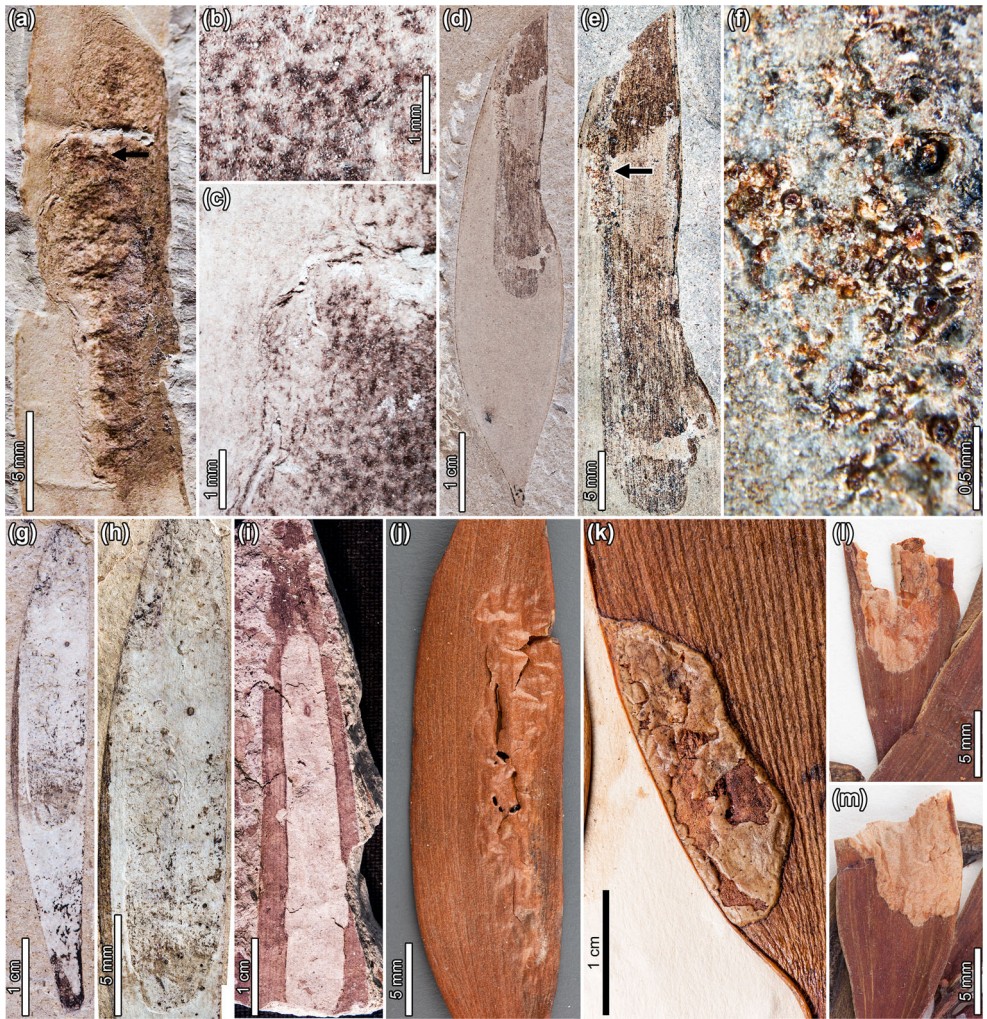

**Fig. 1 Blotch mines on fossil *Agathis* and *Agathis*-like leaves from Patagonia (DT88) and extant *Agathis* leaves. a** *Frondicuniculum flexuosum* blotch mine with wavy margins on *A. immortalis*. Arrow expands to **c** (Danian, locality PL2, holotype MPEF-Pb 5970). **b** Detail of spheroidal frass surrounded by amorphous matter in counterpart of **a**. **c** Detail of wavy, wrinkled margin in **a**. **d** *Frondicuniculum lineacurvum* blotch mine with smooth margins and densely packed frass on *A. zamunerae* (early Eocene, LH06, and holotype MPEF-Pb 6336). **e** Close-up of mine in **d**. Arrow expands to **f**. **f** Spheroidal frass pellets coated or replaced with amber in **e**. **g** *Frondicuniculum lineacurvum* blotch mine spanning the width of the leaf on *A. zamunerae* (middle Eocene, RP, USNM 545226). **h** Close-up of mine in **g**. **i** Elongate, ellipsoidal blotch mine along the central axis of a cf. *Agathis* sp. leaf (Maastrichtian, LefL, MPEF-Pb 9839). **j** Elongate blotch mine near the leaf margin of *A. atropurpurea*, collected in leaf litter from Mount Bartle Frere (Queensland, Australia). **k** Blotch mine along the leaf margin on *A. borneensis* (Brunei, SING 0091231). **l** Partial blotch mine near the leaf margin on *A. microstachya* (Queensland, Australia, K 000553298). **m** Partial blotch mine along the margin of *A. microstachya* (Queensland, Australia, K 000553298).

present at contact between mine margins and surrounding leaf tissue. Individual specimen descriptions of holotype and paratypes provided in Supplementary Note 1.

**Occurrence**. Huitrera Formation, Laguna del Hunco (early Eocene, Chubut Province, Argentina) and Río Pichileufú (middle Eocene, Río Negro Province, Argentina), on host plant *Agathis zamunerae* Wilf.

**Repositories**. Museo Paleontológico Egidio Feruglio, Trelew, Chubut, Argentina (MPEF-Pb), and Smithsonian Institution, National Museum of Natural History (USNM).

*Frondicuniculum flexuosum* ichnosp. nov.

**Etymology**. Classical Latin: *flexuosus –a –um*, full of winding turns, bent, or crooked.
**Holotype**. MPEF-Pb 5970 (Fig. 1a–c and Supplementary Fig. 2a–c), Palacio de los Loros 2 (PL2)[42], Salamanca Formation, early Paleocene, Chubut Province, Argentina.

**Paratypes**. MPEF-Pb 5960 (Supplementary Fig. 2d, e), MPEF-Pb 6007 (Supplementary Fig. 2f, g), MPEF-Pb 6001 (Supplementary Fig. 2h–j), all from the PL2 locality.
**Diagnosis**. As for the genus, with wavy mine margins.
**Description**. An elongate-ellipsoidal blotch mine with gentle to strongly undulatory margins having a raised, wrinkly appearance. Mine positioned along leaf margin, long axis of the mine parallel to leaf veins. Mine dimensions 11.4–35.2 mm long by 1.2–9.4 mm wide. Frass, when present, composed of spheroidal pellets measuring ca. 0.1 mm in diameter and surrounded by smaller fragments of amorphous frass. Frass distributed throughout mine or positioned laterally near one margin of the mine. Mine margins 0.2–8.0 mm wide. Individual specimen descriptions of holotype and paratypes provided in Supplementary Note 1.
**Occurrence**. Palacio de los Loros 2 locality; Salamanca Formation, early Paleocene; Chubut Province, Argentina, on host

plant *Agathis immortalis* Escapa, Iglesias, Wilf, Catalano, Caraballo et Cúneo.

**Repository**. Museo Paleontológico Egidio Feruglio, Trelew, Chubut, Argentina (MPEF-Pb).

**Remarks**. For clarity, we note that the new zoological typifications and identifications assigned here refer only to the insect-damaged areas (i.e., trace fossils) of the cited fossil material, which often has separate botanical typification and identification under the same repository numbers as defined by Wilf et al.[22] for *Agathis zamunerae* and Escapa et al.[7] for *Agathis immortalis*. Morphologically similar, elongate-ellipsoidal blotch mines are associated with *Agathis* at PL2 (early Paleocene, 4 specimens), LH (early Eocene, 2 specimens), and RP (early/middle Eocene, 1 specimen), as well as Cretaceous cf. *Agathis* (see next paragraph), with minor differences in their margin structure. *Frondiculuculum flexuosum* mines have undulatory, wrinkled margins (Fig. 1a, c, and Supplementary Fig. 2a–j), whereas *F. lineacurvum* mines (Fig. 1d–h and Supplementary Figs. 3a–f, 4a, b) have smooth, gently curving margins. However, the overall shape, position on leaves, frass characters, and persistence through ca. 18 myr on the same host genus from the same region suggest that the mines were made by similar, probably closely related leaf-mining insects.

A blotch mine positioned along the central axis of a cf. *Agathis* leaf from the Maastrichtian Lefipán Formation is characterized by an elongate-ellipsoidal shape with its long axis parallel to the leaf veins and smooth, gently curved margins (Fig. 1i and Supplementary Fig. 1a–c). The mine lacks frass, which may be a preservational effect, and otherwise could be the same as *Frondiculuculum lineacurvum*. Because of the preservation and because there is only one specimen, or possibly two (Supplementary Fig. 1d), we did not assign a formal name to this specimen. However, the overall similarity of Cretaceous and Paleocene blotch mines on *Agathis* (elongate-ellipsoidal shape, smooth margins, distorted leaf veins) is noteworthy as the first likely evidence of a Cretaceous-Paleogene (K-Pg) boundary crossing leaf-mine association on closely related plants. Until now, no evidence has been found of surviving K-Pg leaf-mine associations within regional Maastrichtian and Danian floras anywhere in the world[46,49,50].

Another probable blotch-mine type from LH and RP (Eocene) has a linear trajectory and is oriented parallel to the leaf veins, exhibiting breached epidermal tissue (DT251; Fig. 2a, b and see Supplementary Note 1 for detailed descriptions). The putative mines have a similar appearance to slot feeding characterized by elongate holes, although their smooth, gently curving margins suggest a leaf-mining origin. Some of these mine-like structures are flanked by flaps of epidermal tissue, attributable to breaching of the tissue due to environmental factors such as in vivo abrasion (Fig. 2a). The margins along the field of damage are smooth and sometimes influenced by leaf veins. We found similar damage as possible mines on modern Australian *Agathis robusta* (Fig. 2c and Supplementary Fig. 5m, Supplementary Note 1), featuring an elongate-ellipsoidal shape oriented parallel to the leaf veins. Like the fossils, the epidermal tissue of the extant mines is often breached (Fig. 2c and Supplementary Fig. 5m), leaving flaps of unconsumed tissue surrounded by a thin, darkened rim of reaction tissue.

Only two extant leaf-mining insects have been documented in association with *Agathis* (Supplementary Data 1), both on *A. australis* of New Zealand, although their mines are not similar to the fossils. Larvae of the leaf blotch-miner moth *Parectopa leucocyma* (Lepidoptera: Gracillariidae) initially form small blotch mines that transition to linear epidermal mines and then galls[39]. *Microlamia pygmaea*, a longhorn beetle (Coleoptera: Cerambycidae), mines dead leaves on fallen branches[51]. In our survey of extant *Agathis*, we found numerous examples of blotch mines similar to our fossils on six host species that span much of the modern range of the genus (Fig. 1j–m, Supplementary Fig. 5a–f, Supplementary Note 1). The extant blotch mines, previously undocumented to our knowledge, are typically elongate-ellipsoidal and exhibit many similarities to the fossils, suggesting geologically long-term behaviors with origins in the late Mesozoic and early Cenozoic. Most mine trajectories occur along the leaf margins (Fig. 1j–m), although some course along the central axes of leaves (Supplementary Fig. 5c). The long axes of the extant blotch mines are parallel to leaf venation (Fig. 1j–m) as in the fossils. Margins of the mines are smooth to wavy.

In order to assess potential convergence of leaf mine morphologies associated with related conifers that have similar leaf architecture to *Agathis*, we also compared extant mines on *Araucaria* (Araucariaceae) and members of the Podocarpaceae family, including *Nageia*, *Afrocarpus*, *Sundacarpus*, and *Podocarpus*. *Araucarivora gentilii* Hodges (Elachistidae) caterpillars mine leaves of *Araucaria araucana* in Argentina and Chile[52]. Mines begin with a short serpentine phase and then expand into a raised, circular to polylobate blotch mine. A circular exit hole is typically positioned near the margin of the blotch mine. The fossil *Agathis* blotch mines (Fig. 1a–i and Supplementary Figs. 1–4) differ in that they are elongate and lack a serpentine phase. We did not find any other blotch mine morphologies on *Araucaria* herbarium specimens throughout the range of the genus (parts of South America and Australasia). Three leaf-mining taxa have been described on *Podocarpus*. *Phyllocnistis podocarpa* (Lepidoptera: Gracillariidae) larvae mine *Podocarpus macrophyllus* leaves in Japan, creating serpentine mines with overlapping paths that often form into a blotch, although their zigzag frass trail is distinct from the fossil *Agathis* blotch mines[53]. In New Zealand, *Podocarpus totara* hosts two leaf miners, including *Chrysorthenches polita* (Lepidoptera: Glyphipterigidae), whose mines have not been described[54], and *Peristoreus flavitarsis* (Coleoptera: Curculionidae)[55]. The mines of *Peristoreus flavitarsis* are a possible extant analog to the fossil *Agathis* mines, in addition to similar mines we found on extant *Agathis* (Fig. 1j–m and Supplementary Fig. 5). The mines are full depth and typically span the width of the leaf. Frass pellets are often deposited along portions of the mine margin at the edge of the leaf. Individual larvae make mines on multiple leaves. Before pupating in the soil or litter, the larva chews a circular hole through the epidermis on the abaxial side of the leaf[55]. We found other putative blotch mines on herbarium sheets of *Podocarpus* with similar morphologies to those on extant and fossil *Agathis*, including on *Podocarpus ingensis* from Bolivia, *Podocarpus oleifolius* from Colombia, and *Podocarpus urbanii* from Jamaica. We did not find any comparable mines on herbarium specimens of *Afrocarpus*, *Nageia*, or *Sundacarpus*.

Leaf mines have been recognized on other broadleaved, parallel-veined gymnosperms from the Mesozoic, although *Frondiculuculum* is distinguished from all the Mesozoic examples by its blotch morphology and lack of a serpentine phase. Similar to our Cretaceous specimens (Fig. 2d, e, and Supplementary Fig. 1e, f), unnamed mines on the voltzialean conifer *Heidiphyllum elongatum*, from the Late Triassic (Carnian) of the Karoo Basin in South Africa[56,57], exhibit an elongate, parallel-sided, rectilinear path with spheroidal pellets often deposited in an approximate meniscate pattern. *Triassohyponomus dinmorensis* mines, also on *H. elongatum* but from the Blackstone Formation (Middle Triassic) in Australia[58], are serpentine and have an

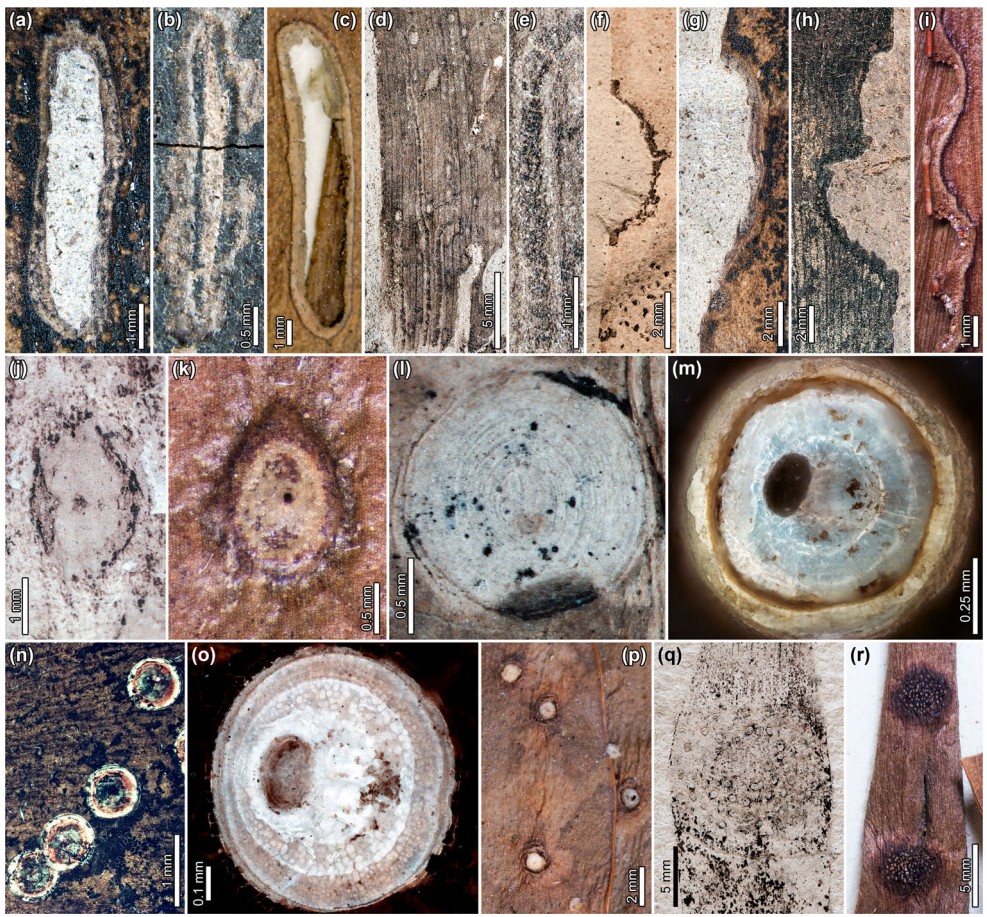

**Fig. 2 External foliage feeding, blotch and serpentine mining, galling, and possible armored scale insect remains (Diaspididae) on fossil and extant *Agathis*. a** Putative blotch mine, or slot feeding, characterized by parallel sides and flaps of necrotic tissue on *A. zamunerae* (early Eocene, LH13, MPEF-Pb 6361). **b** Elongate blotch mine with breached epidermal tissue and thickened reaction rim on *A. zamunerae* (early middle Eocene, RP, USNM 545227). **c** Blotch mine, or possible slot feeding, flanked by flap of epidermal tissue on *A. robusta* (Queensland, Australia, (*A.K. Irvine 00417* (A)). **d** Linear serpentine mines following leaf venation on cf. *Agathis* (latest Cretaceous, DT139; MPEF-Pb 9836). **e** Detail of frass trail in **d**. **f** Semicircular excision into the leaf margin on *A. immortalis* (Danian, PL2, MPEF-Pb 6091). **g** Shallow excision into the leaf margin with vein stringers on *A. zamunerae* (early Eocene, LH13, MPEF-Pb 6361). **h** Two adjacent excisions into the leaf margin on *A. zamunerae* (early/middle Eocene, RP, BAR 5002). **i** Two adjacent excisions into the leaf margin of *A. moorei* (New Caledonia, E 00106192). **j** Ellipsoidal gall with thickened walls surrounding unthickened epidermal tissue on *A. immortalis* (Danian, PL2, MPEF-Pb 9767). **k** Ellipsoidal gall with circular exit hole on *A. ovata* (New Caledonia, E 00399687). **l** Possible armored scale cover (Diaspididae) with concentric growth rings on *A. immortalis* (Danian, PL2, MPEF-Pb 5996). **m** Possible diaspidid scale cover on *A. zamunerae*, under epifluorescence (early Eocene, LH27, MPEF-Pb 6383). **n** Possible diaspidid scale covers on *A. zamunerae*, under epifluorescence (early/middle Eocene, RP, USNM 545228). **o** Possible diaspidid scale cover with concentric growth rings indicating two larval and an adult growth stage on *A. zamunerae*, under epifluorescence (middle Eocene, RP, USNM 545228). **p** Diaspidid scale insects that induced pit galls on *A. macrophylla* (Fiji, GH 01153259). **q** Rust fungus (Pucciniales) with aecia on a circular spot on *A. zamunerae* (early Eocene, LH06, MPEF-Pb 6303). **r** Kauri rust (*Aecidium fragiforme*) on *A. macrophylla* (Vanuatu, *S.F. Kajewski 282* (K)).

extensive, tightly sinusoidal to meniscate pattern. *Fossafolia offae* on *Liaoningocladus boii* from the Early Cretaceous Yixian Formation of northeast China begins as a serpentine mine with an intestiniform frass trail and transitions to a blotch phase[59]. The blotch mines on Patagonian fossil and living Old World *Agathis* therefore appear to represent part of a suite of leaf mining insects of uncertain interrelationships that has colonized parallel-veined, broadleaved seed plants since the Mesozoic, but which nevertheless are distinct from the Mesozoic examples in producing the blotch-mine morphology with no serpentine phase.

**Additional damage diversity.** In addition to leaf mines, fossil and extant *Agathis* are associated with a variety of other insect feeding types (Table 1), which we sketch here while details are being prepared separately. External foliage feeding includes small circular holes (DT1, DT2; Table 1), semicircular excisions into leaf

margins (DT12; Fig. 2f–h), and swaths of surface feeding (DT29). A similar spectrum of damage is found on extant *Agathis* throughout its range (Fig. 2i). However, many types of external foliage-feeding damage can be made by a variety of insects with mandibulate mouthparts across several taxonomic orders[60], and their presence at multiple fossil and modern sites does not necessarily indicate that the same suite of closely-related insect groups produced the damage.

Four gall DTs are associated with fossil *Agathis* in Patagonia, including nondiagnostic, dark circular galls (DT32; Lef and LH), circular galls with a nonhardened center surrounded by a thickened outer rim (DT11; PL2), and columnar galls (DT116; PL2). Moreover, at PL2, *A. immortalis* is associated with ellipsoidal galls bearing a thickened outer wall surrounding epidermal tissue with files of cells. The center of each gall is marked by a circular dot representing the central chamber or exit hole (Fig. 2j). The fossils

**Table 1 Insect damage types on fossil and extant *Agathis*.**

| Insect damage | DT | Lef | PL2 | LH | RP | Extant *Agathis* |
|---|---|---|---|---|---|---|
| **Hole feeding** | | | | | | |
| Circular, <1 mm diam. | 1 | X | X | X | | X |
| Circular, 1–5 mm diam. | 2 | | X | | | X |
| Parallel sided slots | 8 | | | X | | X |
| **Margin feeding** | | | | | | |
| Arcuate excision | 12 | X | X | X | X | X |
| **Skeletonization** | | | | | | |
| Interveinal tissue removed, reaction rim | 17 | | | | X | |
| **Surface feeding** | | | | | | |
| Surface abrasion, weak reaction rim | 29 | X | X | X | | X |
| Polylobate abrasion, strong reaction rim | 30 | | | X | X | |
| **Piercing and sucking** | | | | | | |
| Circular punctures, <2 mm diam. | 46 | X | X | X | | X |
| Scale cover, concentric growth rings | 86 | | X | X | X | X |
| **Oviposition** | | | | | | |
| Ovoidal scar, prominent reaction rim | 101 | X | | | | |
| **Mining** | | | | | | |
| Elongate-ellipsoidal blotch | 88 | X | X | X | X | X |
| Serpentine mine, follows parallel veins | 139 | X | | | | |
| Linear blotch, breached epidermal tissue | 251 | | | X | X | X |
| **Galling** | | | | | | |
| Unhardened central chamber, thickened outer rim | 11 | | X | | | X |
| Nondiagnostic, dark, circular | 32 | X | | X | | X |
| Epidermal center, hardened outer walls | 115 | | X | | | X |
| Columnar gall | 116 | | X | | | |
| **Pathogen** | | | | | | |
| Circular epiphyllous rust fungus with somewhat concentric aecia | 66 | | | X | | X |

*DT* damage type, *Lef* Lefipán localities (Maastrichtian), *PL2* Palacio de los Loros 2 (early Paleocene), *LH* Laguna del Hunco (early Eocene), *RP* Río Pichileufú (middle Eocene). Functional feeding groups are bolded.

resemble undescribed ellipsoidal galls on *A. ovata* from New Caledonia, which are characterized by a raised rim of thickened tissue surrounding a flat, epidermal surface with a circular exit hole (Fig. 2k). The only previously documented galling insect on extant *Agathis* is *Conifericoccus agathidis* (Hemiptera: Margarodidae), the kauri coccid, whose second-instar nymphs induce blister galls that have caused extensive defoliation of *Agathis robusta* in Australia[61]. Nevertheless, we found abundant and diverse gall morphologies on extant *Agathis* (Fig. 2k and Table 1).

Possible covers of female armored scale insects (Diaspididae) occur on *Agathis* at PL2 (Fig. 2l), LH (Fig. 2m), and RP (Fig. 2n, o). At PL2, the covers are found on leaves and a cone scale[7]. The dorsal covers are characterized by concentric growth rings made during the construction of the cover through two instars and an adult phase (Fig. 2l–o). Ventral covers surround the dorsal covers (Fig. 2m) and, in some cases, appear to be deeply set in the leaf tissue and leave a columnar or circular pit when detached. We caution that other interpretations of these structures remain possible because some of their features are not present on extant diaspidid covers (off-center indent or hole on dorsal covers) or are atypical (ventral cover structure surrounding dorsal cover; only adult female covers are present). The possible scale covers, including the off-center indent, are very similar to structures assigned to Diaspididae from the Late Cretaceous of New Zealand and Australia[62], and comparable scale covers are associated with angiosperm leaves from the Eocene of Germany[63] and Miocene of New Zealand[64]. On extant *Agathis*, diaspidid scales previously have been documented on three species in New Zealand and Australia (Supplementary Data 1). We found diaspidids on *Agathis* herbarium specimens from New Caledonia and Fiji, including an unidentified diaspidid species that induced pit galls on *A. macrophylla* from Fiji (Fig. 2p).

A probable rust fungus (Pucciniales), characterized by rings of circular to oval aecia on a circular gall, is associated with *Agathis zamunerae* at LH (Fig. 2q). Two species of rust fungi in the genus *Aecidium* parasitize extant *Agathis*: *Aecidium fragiforme* from Oceania and Malesia and *Aecidium balansae* in New Caledonia (Fig. 2r)[40,65]. *Aecidium* on extant *Agathis* produces galls covered in yellow aecia (Fig. 2p). The very similar morphologies of the fossil and extant rust on *Agathis* suggest long-term, persistent associations reaching back to at least the early Eocene.

## Discussion

The persistence of multiple, nearly identical insect herbivore and fungal associations for tens of millions of years on the same host genus, *Agathis* (Table 1), provides insights into the evolution of insect-herbivore component communities in deep time[66–68]. The first occurrences of some of these associations were on the early Paleocene stem-group species *A. immortalis* and continued onto the early Eocene crown group *A. zamunerae* from ca. 12 million years later, suggesting an evolving co-association dating to the earliest history of the host genus[67]. For example, the mine margins of *Frondiculiculum flexuosum* on *A. immortalis* are wavy, but the margins of *F. lineacurvum* on *A. zamunerae* are linear to gently curving, suggesting evolutionary changes in leaf-miner behavior from the Paleocene to the Eocene. Besides minor morphological changes, however, the component communities of the Paleocene stem group and Eocene crown-group *Agathis* species remained remarkably similar (Table 1). The presence of crown-group *Agathis* in Gondwanan rainforests by the early Eocene[22] and the established pattern of rainforest biome tracking among the survivor taxa from the fossil Patagonian floras[5,69] provide evidence for niche conservatism[70,71]. Extant *Agathis* has

accessory xylem tissues adjacent to leaf veins[72] that can collapse during drought conditions[73], highlighting the importance of everwet rainforest environments for the genus. The inferred presence of rainforest environments throughout the history of Agathis[5,22] may have provided stable conditions for insect and fungal associations to establish geologically long-term, conservative relationships with their host genus, even as the rainforest habitats relocated across vast distances in response to global change.

Another possible explanation for reoccurrences of similar associations on Agathis through time is the convergence of insect and fungal damage morphologies and tissue usage. In this scenario, some or all the culprit lineages that made the damage on Agathis in ancient South America went extinct, even as the host persisted, and were replaced by unrelated but ecomorphically convergent lineages that recolonized the same host at a later time. The presumed stability of environmental pressures on potential herbivores as a result of habitat tracking by Agathis may have led to a convergence in damage type morphologies and diversity through time. In addition, the leaf morphology of Agathis has remained relatively unchanged since at least the latest Cretaceous[7], and leaf mining behavior in particular can be influenced by leaf architecture[74]. Podocarpus was the only Gondwanan conifer genus that we found with similar blotch mines to Agathis, which may represent an example of convergence of leaf mine morphology as a result of similar leaf structure. However, we did not find analogs on Podocarpus to other distinct associations on fossil Agathis from Patagonia, such as the second blotch mine type, pit galls, and a distinctive rust fungus. Thus, although Podocarpus hosts morphologically similar blotch mines to Agathis, the full suite of associations on fossil Agathis presented here persists only on living Agathis.

The convergence scenario is possible and in all likelihood occurred for at least some of the fossil examples we report here that also occur on living Agathis. Because the modern and ancient culprits discovered here, are by definition, largely unknown, it is not yet possible to distinguish definitively between lineage persistence and convergence, and we present this contrast as a question for future work. However, persistence is more parsimonious and plausible than convergence on first principles, given that the host lineage clearly survived and maintains a remarkably similar suite of biotic associations today as in the deep past that is not known to occur on any other living genus.

In summary, we found a similar suite of insect damage on Cretaceous, Paleocene, and Eocene Agathis fossils from Patagonia as well as on extant Agathis throughout its modern range (Table 1). This persistent damage included leaf mines as well as external foliage feeding, galling, possible diaspidid scale-insect covers, and a rust fungus. The repeated occurrences of compositionally and morphologically similar types of herbivory on Agathis throughout its history suggest, at least, that functionally conservative ecological guilds of insects and fungi have persisted on Agathis since the end of the Mesozoic or early Cenozoic. A further possibility, and a likely one because of the demonstrated survival of the host genus, is that living Agathis component communities include the legacy of long-term, ecologically conservative lineage associations that originated in Mesozoic Gondwana, survived the end-Cretaceous extinction, and tracked the host–plant genus Agathis as its crown lineage evolved. In this scenario, the host and its communities then shifted their ranges together slowly but massively in response to major plate movements and environmental changes. These associations persist today at great temporal and spatial distances in Australasia and Southeast Asia, but they are now critically threatened by human impacts, including logging[38] and climate change[75]. Most of the living culprits of Agathis herbivory that we observed are unknown

to the best of our knowledge, suggesting potential for evolutionary and biogeographic study of novel associations.

## Methods

**Fossil *Agathis* data collection.** We compared insect feeding damage on Patagonian fossil Agathis leaves, branches, and cone scales (Supplementary Data 2) from well-studied localities of the latest Cretaceous (cf. Agathis sp., Lefipán Formation, localities LefE, LefL and, LefW, 10 specimens)[45,46], early Paleocene (A. immortalis, Salamanca Formation, locality PL2, 319 specimens)[7,42–44], early Eocene (A. zamunerae, LH, 121 specimens)[11,21,22], and middle Eocene (A. zamunerae, RP, 32 specimens)[11,21,22] with damage on extant Agathis species. The Lefipán localities and LH are located in northwest Chubut, Argentina; PL2 is in the western San Jorge Basin near Sarmiento in southern Chubut; and RP is located in western Río Negro, Argentina. The fossil plant collections examined from Lefipán, PL2, and LH are curated at the Museo Paleontológico Egidio Feruglio (MPEF-Pb) in Trelew, Chubut. The RP collections are curated at the Museo de Paleontológico de Bariloche (BAR), San Carlos de Bariloche, Río Negro, and E.W. Berry's historical Río Pichileufú collection[48] is housed at the Smithsonian Institution, National Museum of Natural History (USNM), Washington, D.C., United States. We assigned damage-type numbers (DTs) to all insect and fungal damage using the standard reference[76] and subsequent addenda. Due to the small sample sizes of rare fossil conifer-leaf mines, we did not attempt quantitative analyses comparing time periods.

**Fossil locality information.** The Lefipán formation was deposited in the Maastrichtian to early Danian in a tide-dominated delta environment[45]. The macrofloras studied here are from the terminal Maastrichtian portion of the formation (67–66 Ma), based on biostratigraphy of marine invertebrates[45], dinoflagellates, and palynomorphs[77,78]. The three fossil plant localites (LefE, LefL, and LefW) are located in close proximity in the San Ramón section of the Lefipán Formation[45], as previously detailed[46]. The cf. Agathis leaves from the Lefipán Formation display typical Agathis characters (symmetrical blade, lanceolate shape, parallel venation, short constricted petiole), but no reproductive parts have been found, and the specimens lack cuticular remains[7]. Nevertheless, they are only ca. 2–3 m.y. older than the Danian Agathis immortalis fossils, come from the same region and general depositional setting, and quite possibly represent A. immortalis or a close relative[7].

Early Danian A. immortalis fossils were collected from locality Palacio de los Loros 2 (PL2) in the estuarine Salamanca Formation[44]. The strata at PL2[42] were deposited during geomagnetic polarity chron C28n (64.67–63.49 Ma)[43,44,79]. Specimens of A. immortalis include leaves, both isolated and on leafy branches, and many cone scales, winged seeds, and pollen cones with in situ Dilwynites pollen, including juvenile pollen cones attached to a leafy branch[7].

Eocene Agathis zamunerae fossils come from Laguna del Hunco (12 quarries) and Río Pichileufú (specimens are from quarries RP3, near RP1, or imprecisely known localities from older collections), both of these caldera-lake deposits in La Huitrera Formation[22]. At Laguna del Hunco, $^{40}$Ar/$^{39}$Ar dating of sanidine crystals in a tuff within a fossiliferous layer yielded an age of $52.22 \pm 0.22$ Ma[10,11,21], and $^{40}$Ar/$^{39}$Ar dating of tuffs located directly over the fossiliferous layers at Río Pichileufú provided an age of $47.74 \pm 0.05$ Ma[11,21].

**Extant Agathis data collection.** We examined herbarium specimens of 15 species of Agathis for insect damage (Supplementary Note 1), including the complete collections of Agathis at the Arnold Arboretum (A) and Gray Herbarium (GH) of the Harvard University Herbaria, Royal Botanic Garden Edinburgh (E), Royal Botanic Gardens Kew (K), United States National Herbarium (US), Australian National Herbarium (CANB), National Herbarium of New South Wales (NSW), and the Singapore Botanic Gardens Herbarium (SING). Field examinations of Agathis were done in the Atherton Tablelands region, Queensland, Australia (A. robusta, A. atropurpurea, and A. microstachya) and on Mt. Kinabalu, Sabah, Malaysian Borneo (A. lenticula, A. borneensis, and A. kinabaluensis). We also examined the digitized (data.rbge.org.uk/search/herbarium) collections of Afrocarpus, Podocarpus, and Sundacarpus (Podocarpaceae) and Araucaria (Araucariaceae) from E and physical collections of Nageia (Podocarpaceae) from E and K to assess convergence in leaf mine morphology with Agathis mines.

**Photography.** We took macrophotographs with a Nikon D90 camera at the repositories listed above and at the Paleobotany Laboratory, Pennsylvania State University. For microphotography, we used a Nikon SMZ 1500 binocular microscope with a Nikon DS-Ri1 camera at the Paleobotany Laboratory, Pennsylvania State University. We integrated images with Nikon NIS Elements v. 3 software, used Photoshop CC 2017 to vertically composite photographs of fossils when necessary, and Adobe Camera Raw Editor to adjust contrast, white balance, etc. on entire images.

**Nomenclatural acts.** This published work and the nomenclatural acts it contains have been registered in ZooBank, the proposed online registration system for the International Code of Zoological Nomenclature (ICZN). The ZooBank LSIDs (Life Science Identifiers) can be resolved and the associated information viewed through

any standard web browser by appending the LSID to the prefix "http://zoobank.org/". The LSIDs for this publication are: 12D2484C-3DEF-4B47-B307-E79EA61E8C4E for the new ichnogenus *Frondicuniculum*; 31540A57-B065-4BFD-8F67-021943A29F43 for the new ichnospecies *F. flexuosum*; and E8C206D0-22F5-4A2F-8183-F4AC51B3DA54 for the new ichnospecies *F. lineacurvum*.

**Reporting summary**. Further information on research design is available in the Nature Research Reporting Summary linked to this article.

## Data availability

All data generated during this study are included in this published article and its supplementary information files. All material cited in this study is housed and permanently available in the repositories named in this article. Full repository information for all fossil specimens studied is listed in Supplementary Data 2.

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

## Acknowledgements
The authors thank E. Pinheiro and two anonymous reviewers for their constructive comments; I. Escapa, P. Puerta, L. Reiner, and E. Ruigómez and many others from Museo Paleontológico Egidio Feruglio and elsewhere for field and laboratory assistance; the staff at E (L. Scott), A and GH (J. Shapiro and E. Wood), K (S. Dawson, A. Farjon), US (E. Gardner, A. Clark, I. Lin, K. Rankin, R. Russell), CANB (B. Lepschi, C. Cargill), NSW (L. L. Lee and L. Murray), SING (N. Karim, S. Lee), USNM (F. Marsh and J. Wingerath), and CMNH (P. Fox and D. Su) for help with their collections; S. Bellgard, T. Bralower, D. Hughes, Y. Imada, N. Martin, M. Padamsee, and M. Patzkowsky for discussions; R. Kooyman for fieldwork with PW in Queensland and Sabah and related collaboration; and D. Crayn for further assistance with Australian material. This study was supported by grants to M.P.D. from the CIC/Smithsonian Fellowship, the Evolving Earth Foundation, the Geological Society of America, Sigma Xi, the Paleontological Society and the P. D. Krynine Memorial Fund of the Penn State Department of Geosciences; and to P.W., A.I., and N.R.C. from NSF awards DEB-0345750, DEB-0919071, DEB-1556666, and EAR-1925755. This research partly fulfilled requirements for a PhD degree in Geosciences from Pennsylvania State University by MPD.

## Author contributions
M.P.D., P.W., C.C.L., and A.I. designed research. A.I., P.W., N.R.C., and M.P.D. did the fieldwork. M.P.D., A.I., C.C.L., and P.W. collected the DT data. N.R.C. led research on the Lefipán flora, A.I. on the Paleocene floras, and P.W. on the Eocene floras. M.P.D. analyzed the data and wrote the manuscript with comments from all other authors.

## Competing interests
The authors declare no competing interests.
