## [Peer Review File · Communications Biology]

Reviewers' comments:

Reviewer #1 (Remarks to the Author):

Dear Dr Donovan

I really liked the idea to track plant-insect and fungal associations in *Agathis* through time. However, the presentation of the results are not consistent with the goal and conclusions. I know the journal has a limited number of words, so, you don't have much space to explain, but even so, I think you could present in a less descriptive and more "ecological" way. You did a tremendous work! The audience should see a general view in the main text, not just in the supplementary material. You can see my suggestion in the attached file.

Reviewer #2 (Remarks to the Author):

Summary: This short paper reports a particular insect damage type (DT88) along with other damage on fossil leaves of the conifer *Agathis* from South America. Highly similar damage is found on living leaves of *Agathis*. It is proposed that the similar suite of damage on living and fossil *Agathis* suggests long term persistence of ecological guilds and possible host tracking of the genus across time (since the Mesozoic) and space (across Gondwana). The reported insect damage type has previously been described (in 2007) but is formally described as two new ichnospecies in the present paper.

Overall impression of the work: The manuscript is well written. However, it is a very technical paper and I wonder whether it is of much relevance to a wider audience. Much of the paper and the supplementary material is devoted to the description of a particular damage type in a conifer that belongs to a group that existed at least since the early Mesozoic. The Discussion section is extremely short and does not place the new findings in a broader context. I would have expected a thorough discussion of why the damage type in the fossil leaves (caused by unknown insects) should have been produced by the same insects in the modern leaves (it would actually be exciting to know which insects make this damage in the living trees). Further, I would have expected a thorough comparison of modern damage types in *Agathis* with closely related modern groups such as *Podocarpus*, and *Araucaria* (especially since the fossil damage is also reported for an *Agathis* stem group fossil). In the discussion, the wider implications of this report should have been discussed. All this is missing.

Specific comments, with recommendations for addressing each comment:

Lines 25-26: This question cannot be answered by the present paper as the "culprits" are still unknown. See also lines 34-36.

To prove this you would need to offer an in depth discussion about how specific the observed DTs are.

Lines 38, 39: "living-fossil" association – this is a nice wordplay (I assume you do not mean "living fossil") but contradictory and potentially difficult to follow: A newly discovered "living-fossil" association would imply that we know that both the fossil and modern culprits were the same. From your text it is clear that we do not know.

Lines 158-160: Perhaps yes, but I wonder whether the specific leaf architecture of the podocarp-Araucaria-*Agathis* clade is prone to strongly convergent patterns of leaf damage. And how is it in grasses?

In order to back your idea of a long-term association you need to demonstrate the specificity of the reported mining and that only a particular group of insects produces such mining.

Lines 188-195: Here it would be necessary to investigate closely related conifer genera with very similar leaf architecture as well.

Lines 196-208: You say that the unknown insects have colonized "parallel-veined, broadleaved seed plants since the Mesozoic". This is much more modest (and perhaps more sensible based on what we know today) than the claim of a specific host (*Agathis*) tracking since the Cretaceous of a specific insect.

Lines 215-219: How then would you rule out that the blotch mining is done by different groups of insects?

Line 256 ff: Discussion: Here you essentially say that niche conservatism accounts for the long-term associations that originated in the late Cretaceous. I am fine with that but I would like a more in depth discussion of why you think there is sufficient evidence to rule out convergent evolution in these similar types of herbivory, especially in conifer leaves with a high degree of convergent evolution of leaf architecture.

Line 533: Figure 1:

I realized that in the guide to insect damage types from 2007, the very same leaf (your Fig 1, D) was used to illustrate DT88. This made me curious about how common this damage type is and whether it is restricted to particular plant groups/plant functional types.

A simple survey in the electronic herbarium catalogue of Edinburgh (herbarium E) shows that pretty similar blotch DTs are occurring on podocarps. E.g.
https://data.rbge.org.uk/search/herbarium/?specimen_num=434594&cfg=zoom.cfg&filename=E00399714.zip

Are there distinct DTs on closely related *Agathis*, *Podocarpus*, *Araucaria*?

According to molecular dating studies, this group dates back to at least the Triassic/Jurassic. See e.g. Ran et al 2018 (Proc. R. Soc. B 285: 20181012)

Leslie et al 2018 (Am J Bot 105(9): 1531–1544) estimated a clade age for Podocarpaceae + *Araucaria* and agathoids dating back to the early Triassic.

Therefore, I would expect a long period of shared evolutionary history. It would be lovely if you could elaborate on this in the light of your findings and explaining to a broader audience if and how insect damage types are diagnostic for different lineages in this clade.

This might be useful when discussing the possible long-lasting association between a particular insect (even though unknown) and *Agathis*.

Reviewer #3 (Remarks to the Author):

This is an interesting paper and the approach is novel. The have compiled an impressive amount of relevant data as well. The paper is well written and the illustrations and figures are Good and all necessary especially Table 1. I found little editorial problems with the paper. It appears the

authors have taken the data at face value and the authors do not wander too far from that data. This is a clean, tight manuscript and recommend publication. When this is published I will make required reading in my evolution and paleobotany class as an example of what kind of biological research is possible on fossils. The paper was enjoyable to read. Recommend publication with little or no changes.

Passos, 1st June 2020

Dear Dr Donovan

Your paper “Persistent biotic interactions of a Gondwanan conifer from Cretaceous Patagonia to modern Malesia” will be very important for the paleobotany and plant-insect interactions field of study. However, I would like to suggest some modifications.

I really liked the idea to track plant-insect and fungal associations in *Agathis* through time. However, the presentation of the results are not consistent with the goal and conclusions. I know the journal has a limited number of words, so, you don't have much space to explain, but even so, I think you could present in a less descriptive and more “ecological” way. You did a tremendous work! The audience should see a general view in the main text, not just in the supplementary material. I also have a question for the authors:

All the studied fossil are from Argentina. Do you think that represents how the genus did behave in all his geographical range?

Below, I list and discuss my suggestions of changes:

Introduction:

-It would be good to the reader if the paper call the figure with the modern *Agathis* in the introduction. I think would help to have a bigger view of the idea to track *Agathis* through time. For example, line 73, when the paper mention to have seen similar blotches on fossil and extant species, it should call figure with the blotch in the modern example too.

- Line 77: add the reference to say that *Agathis* is dominant today. It's a strong statement.

- What is the manuscript hypothesis? The goal is clear from line 79 to 82, but what is expected?

Results:

- The sections are not adequate to the goal of the paper in my opinion. The title and the introduction make the reader expect a more ecological and numerical result, not just

ichnotaxonomy and the description of the damages. How many fossil were analyzed per period? How many exsiccates?

- The results section does not have a standard presentation. All ichnospecies should show same presentation. For example: remarks for *F. lineacurvum* and Repositories to *F. flexuosum*.

-Line 209, I think this part should be called just “damage diversity”, since the first part was called “systematic paleontology”.

Discussion:

- The paper brings the words “evolution” and “evolutionary” in a too broad sense. You should avoid that.

- I really liked here the idea of “niche conservatism”. This was the hypothesis in the beginning? If yes, you should add this to the introduction.

- I would like to see a clear conclusion in the end of your discussion.

One last comment, you use the term Malesia just in the title. You could add when you mention the Wallace’s Line.

I would like to remind you that I’m not a native English speaker.

I hope I could help you to improve your manuscript.

Sincerely yours,

Esther Pinheiro

esther.rspinheiro@gmail.com

Response to reviewers

COMMSBIO-20-1170-T

Persistent biotic interactions of a Gondwanan conifer from Cretaceous Patagonia to modern Malesia

Communications Biology

Reviewer #1 (Remarks to the Author):

Dear Dr Donovan

“I really liked the idea to track plant-insect and fungal associations in *Agathis* through time. However, the presentation of the results are not consistent with the goal and conclusions. I know the journal has a limited number of words, so, you don't have much space to explain, but even so, I think you could present in a less descriptive and more “ecological” way. You did a tremendous work! The audience should see a general view in the main text, not just in the supplementary material. You can see my suggestion in the attached file.”

We thank Dr. Pinheiro for the helpful and insightful comments, which helped us to significantly improve our paper. To address the comments on the consistency of the presentation of the results with the goals and conclusions, we added additional explanation to the Discussion on the issue of the persistence vs. convergence scenarios (lines 353-382). We also clarified text in the Introduction (lines 25-29, 77-78, 108-109, 124-125) and Discussion to further illuminate the biogeographic, evolutionary, and ecological implications of the study (lines 341-343, 353-382, 388-390; additional details in the responses to Reviewer 2). We provide more detailed explanations on how we addressed ecological issues in our answers to specific comments from Reviewer 1 below.

1. “*All the studied fossils are from Argentina. Do you think that represents how the genus did behave in all his geographical range?*”

***Agathis* fossils have also been found from the late Paleocene to early Miocene of Australia and the late Oligocene–Miocene of New Zealand as discussed in the introduction. We did not include these fossils in our analysis, however, because *Agathis* fossils are rare in Australia and New Zealand, their sample sizes are low and, most critically here, they do not have reported insect damage (an important detail now noted in the manuscript at line 87). It is very possible that the damage types associated with Patagonian fossil *Agathis* may have been more widespread before the breakup of Gondwana, but larger collections are needed to test this. We note that our paper covers the first *Agathis* fossils ever found in South America or anywhere outside Australia and New Zealand, and they certainly indicate the potential for many other new discoveries in other areas.**

Below, I list and discuss my suggestions of changes:

“Introduction:

2. -It would be good to the reader if the paper call the figure with the modern *Agathis* in the introduction. I think would help to have a bigger view of the idea to track *Agathis* through time. For example, line 73, when the paper mention to have seen similar blotches on fossil and extant species, it should call figure with the blotch in the modern example too”

We agree and have added specific figure citations for fossil and extant leaf mines on *Agathis* (line 100-101)

3. “- Line 77: add the reference to say that *Agathis* is dominant today. It’s a strong statement.”

Good idea. We cited Ecroyd (1982) to support *Agathis* dominance today.

4. “- What is the manuscript hypothesis? The goal is clear from line 79 to 82, but what is expected?”

Thank you. This study was performed after the discovery of similar-to-modern component communities on *Agathis* fossils from Patagonia and was discovery-driven, not hypothesis-based. We did not want to add a hypothesis post-hoc, after performing the study, so we chose to frame the study as an exploratory investigation in terms of its original goals.

“Results:

5. - The sections are not adequate to the goal of the paper in my opinion. The title and the introduction make the reader expect a more ecological and numerical result, not just ichnotaxonomy and the description of the damages. How many fossil where analyzed per period? How many exsiccates?”

We’re sorry this information was not clear. However, the number of fossil specimens we analyzed per species is included in the Materials and Methods section. As there stated, we looked at all *Agathis* specimens from the following herbaria: Arnold Arboretum (A) and Gray Herbarium (GH) of the Harvard University Herbaria, Royal Botanic Garden Edinburgh (E), Royal Botanic Gardens Kew (K), United States National Herbarium (US), Australian National Herbarium (CANB), National Herbarium of New South Wales (NSW), and the Singapore Botanic Gardens Herbarium (SING). We now clarify the important detail that we looked at all *Agathis* specimens at those herbaria on line 356. In the last paragraph of the Introduction (lines 111-141), we gave a short overview of our methods, including erecting a new ichnogenus and two new ichnospecies and comparing insect and fungal damage morphologies on fossil and extant *Agathis*. Quantitative analyses would not change our overall findings, and we did not perform any quantitative analyses comparing time periods due to the rarity of the fossil leaf mines. This is stated in the revised manuscript. If there is something else we can do to clarify these issues, please let us know specifically.

With regard to “a more ecological ... result, not just ichnotaxonomy and the description of the damages”, we feel that especially now with additional Discussion text on the issue of persistence vs. convergence as recommended by Reviewer #2 (see below), the deep background for the whole paper in terms of Gondwanan

biogeography, and the remarkable discoveries reported here from both ancient and living *Agathis*, there is a great deal in this paper that is of broad ecological, biogeographic, and evolutionary interest.

6. “- The results section does not have a standard presentation. All ichnospecies should show same presentation. For example: remarks for *F. lineacurvum* and Repositories to *F. flexuosum*.”

Thank you, and we added a “Repository” section for *F. flexuosum* on line 193. Otherwise, Remarks sections are not required by the ICZN, and our remarks section is constructed to discuss both ichnospecies together for efficiency.

7. “-Line 209, I think this part should be called just “damage diversity”, since the first part was called ‘systematic paleontology’.”

We see the point but still prefer to name the section “Additional Damage Diversity” to highlight insect and fungal damage ‘in addition to’ the leaf mines described in the “Systematic Paleontology”, or first part, of the section, which also describes damage diversity (of those leaf mines).

“Discussion:

8. - The paper brings the words “evolution” and “evolutionary” in a too broad sense. You should avoid that.”

Unfortunately, we are not sure what the reviewer means in this comment and thus cannot respond without further guidance.

9. “- I really liked here the idea of “niche conservatism”. This was the hypothesis in the beginning? If yes, you should add this to the introduction.”

We agree that this is an interesting topic, and we have refined the passages about niche conservatism in the revised Discussion to heighten that interest. However, niche conservatism was not the hypothesis at the beginning of the study. As stated earlier, we performed the study after the discovery of similar component communities on Patagonian *Agathis* from the latest Cretaceous to Eocene to Recent, and it was discovery, not hypothesis driven. We did not want to add a hypothesis after completing the data collection and interpretation. However, we did add references that discuss niche conservatism in the fossil record (Prinzing et al., 2017) and Southern Hemisphere floras (Crisp et al., 2009).

10. “- I would like to see a clear conclusion in the end of your discussion.”

Our conclusions are presented in the revised final paragraph of the Discussion section (lines 383-399), which begins, “In Summary...” We summarized our findings, including the two possible explanations for our results (host tracking and convergence, see below), and also highlighted the potential for future biogeographic studies on the novel extant associations presented in this study. We added to the existing paragraph and revised the language in parts of the paragraph to more clearly articulate our conclusions (lines 388-390, 393-394). If there is something else we can do to improve the concluding text sections please let us know.

11. “One last comment, you use the term Malesia just in the title. You could add when you mention the Wallace’s Line.”

We agree and added “in Malesia” in line 91-92.

Reviewer #2 (Remarks to the Author):

“Summary: This short paper reports a particular insect damage type (DT88) along with other damage on fossil leaves of the conifer *Agathis* from South America. Highly similar damage is found on living leaves of *Agathis*. It is proposed that the similar suite of damage on living and fossil *Agathis* suggests long term persistence of ecological guilds and possible host tracking of the genus across time (since the Mesozoic) and space (across Gondwana). The reported insect damage type has previously been described (in 2007) but is formally described as two new ichnospecies in the present paper.”

“Overall impression of the work: The manuscript is well written. However, it is a very technical paper and I wonder whether it is of much relevance to a wider audience. Much of the paper and the supplementary material is devoted to the description of a particular damage type in a conifer that belongs to a group that existed at least since the early Mesozoic. The Discussion section is extremely short and does not place the new findings in a broader context. I would have expected a thorough discussion of why the damage type in the fossil leaves (caused by unknown insects) should have been produced by the same insects in the modern leaves (it would actually be exciting to know which insects make this damage in the living trees). Further, I would have expected a thorough comparison of modern damage types in *Agathis* with closely related modern groups such as *Podocarpus*, and *Araucaria* (especially since the fossil damage is also reported for an *Agathis* stem group fossil). In the discussion, the wider implications of this report should have been discussed. All this is missing.”

We thank Reviewer #2 for the perceptive and constructive comments, which we feel have helped us to improve our paper significantly. In response to the main comments above, we will first provide an overview of our revisions here, and then present our responses to specific comments below.

We revised language in the Abstract (lines 25-29), Introduction (lines 77-78, 107-109), Results (lines 237-261) and Discussion (lines 341-343, 353-382, 388-390) to make clearer the possibility of convergence and persistence as possible scenarios. These concepts were included in the original submission, but we added additional text to clarify the scenarios, stress that both hypotheses open up the possibilities for future work, and give reasons why persistence is overall more likely.

Reviewer #2 made an important point about the possibility of convergence as a cause of similarities in insect damage on *Agathis* from different time intervals, and suggested comparing mines on related conifers with comparable leaf architecture to *Agathis*, including *Araucaria* (Araucariaceae) and *Podocarpus* (Podocarpaceae) and other podocarps (lines 237-261). To explore this possibility, we surveyed ~2800 scanned herbarium sheets for insect damage from the Royal Botanic Garden Edinburgh (E) herbarium catalog. We also compiled relevant literature on leaf-mining insects associated with related conifers. We found that only *Podocarpus* hosted similar leaf mines to *Agathis*, including a blotch mine made by a weevil in

New Zealand and putative blotch mines on herbarium sheets from Bolivia, Colombia, and Jamaica (lines 257-259).

Given our current understanding of the component communities on extant *Agathis*, we cannot unequivocally assign the blotch mines on *Agathis* to an extant leaf-mining lineage. However, none of the related living conifer genera host anything like the full suite of damage associated with fossil *Agathis* (lines 353-382). We acknowledge, and now state clearly, that both persistence and convergence, and probably some of both, are possible explanations for the patterns observed in this study. Nevertheless, we also state that host tracking is the most parsimonious explanation for the reoccurrence of similar suites of damage on *Agathis*, because the host plant survived through the entire studied interval, including the present day, and hosted remarkably similar associations throughout that are simply not found as a suite on other conifers. We added new text to discuss these key points in lines 353-382.

Finally, we expanded the broad-interest Discussion points to accommodate the points Reviewer #2 made, which should now make the broader context requested clearer. We discuss implications for Gondwanan biogeography, possible scenarios for the evolution of insect and fungal associations on plants over geologic time, and connections with modern threatened ecosystems in Australasia and Southeast Asia (lines 333-399).

We feel that the paper is much more interesting as a direct result of these modifications made in response to Reviewer #2's comments.

“Specific comments, with recommendations for addressing each comment:

1. Lines 25-26: This question cannot be answered by the present paper as the “culprits” are still unknown. See also lines 34-36.” “To prove this you would need to offer an in depth discussion about how specific the observed DTs are.”

We agree that as written, lines 25-26 (“**However, it is unknown whether insect herbivore and fungal communities tracked these host plants across time and space**”) cannot be adequately answered given our current knowledge. Therefore, we changed lines 27-29 to “**However, it is unknown whether dependent ecological guilds or lineages of associated insects and fungi persisted on Gondwanan host plants like *Agathis* through time and space**”. The revised sentence acknowledges the two main scenarios we discuss to explain the patterns of similar plant-insect associations from the latest Cretaceous to middle Eocene of Patagonia and modern Australasia and Southeast Asia: convergence vs. persistence. The revised sentence highlights the possibility of functionally conserved, ecological associations (different lineages converging on similar feeding strategies as guilds) or persistence via host tracking by specific lineages of insects and fungi. Line 34-36 says “**The similar suite of damage on extant and fossil *Agathis* suggests long-term persistence of ecological guilds and possibly the component communities associated with *Agathis* since the late Mesozoic**”. Ecological guilds encompass groups of species that use a resource in similar ways, but are not necessarily taxonomically related, which is consistent with our results. The persistence of component communities on *Agathis*, which implies tracking of the genus by taxonomically-related insects, is included as a second, but not mutually exclusive possibility.”

2. “Lines 38, 39: “living-fossil” association – this is a nice wordplay (I assume you do not mean “living fossil”) but contradictory and potentially difficult to follow: A newly discovered “living-fossil” association would imply that we know that both the fossil and modern culprits were the same. From your text it is clear that we do not know.”

We agree and changed “living-fossil association” to “living association” for accuracy and clarity.

3. “Lines 158-160: Perhaps yes, but I wonder whether the specific leaf architecture of the podocarp-Araucaria-Agathis clade is prone to strongly convergent patterns of leaf damage. And how is it in grasses?”

In order to back your idea of a long-term association you need to demonstrate the specificity of the reported mining and that only a particular group of insects produces such mining.”

We appreciate this question and comment, which prompted us to more thoroughly explore the possibility of convergence of leaf mine morphologies on related conifers with similar leaf architectures to *Agathis*. We added a discussion of leaf mines found on members of Podocarpaceae (*Podocarpus*, *Afrocarpus*, *Nageia*, and *Sundacarpus*) and *Araucaria* (Araucariaceae) in Lines 237-261, including both previously documented mines and mines we found on herbarium sheets. *Podocarpus* was the only conifer genus that hosted similar blotch mines, suggesting possible convergence of leaf mine morphologies on a conifer with similar leaf architecture to *Agathis*. As discussed in the paper and clarified in our revisions (lines 25-29, 77-78, 107-109, 237-261, 341-343, 353-382, 388-390), both host tracking and convergence are possible and non-exclusive explanations for the persistence of similar insect damage types associated with *Agathis* throughout its history. Although we cannot unequivocally assign the *Agathis* blotch mines to a known extant leaf-mining lineage, the recurrence of similar mines on a single host plant genus, *Agathis*, since the latest Cretaceous (the entire history of the genus) is the common variable that ties these leaf-mines together. Grasses host a variety of leaf-mining morphotypes, including serpentine and blotch mines, most commonly made by dipteran larvae with more delicate mines and distinct frass trails that alternate between sides of the mine, typical of Agromyzidae.

4. “Lines 188-195: Here it would be necessary to investigate closely related conifer genera with very similar leaf architecture as well.”

Thank you for this suggestion. We added a discussion of blotch mines on related conifer genera in Lines 237-261, as discussed in the previous comment.

5. “Lines 196-208: You say that the unknown insects have colonized “parallel-veined, broadleaved seed plants since the Mesozoic”. This is much more modest (and perhaps more sensible based on what we know today) than the claim of a specific host (*Agathis*) tracking since the Cretaceous of a specific insect.”

The morphologies of mines on earlier Mesozoic broadleaved conifers differ significantly from the blotch mines that occur for ca. 18 million years on fossil (and living) *Agathis*, as detailed in lines 263-277 and now summarized in lines 276-277.

We appreciate this comment, which prompted us to highlight those key differences. As discussed above, the consistent occurrences of blotch mines and other associations could be a result of persistence or convergence, both potentially driven by presumed environmental stability caused by habitat tracking by *Agathis*.

6. “Lines 215-219: How then would you rule out that the blotch mining is done by different groups of insects?”

Thank you for the question. Multiple insect orders are capable of making similar external foliage feeding damage, such as margin feeding and hole feeding, so that the identities of damage producers are typically not discernible in fossils. Leaf mines and many galls, however, preserve a greater range of morphological and behavioral information as tracemakers than most forms of external foliage feeding. Distinctive features of leaf mines include some combination of leaf mine size and shape, frass trail characteristics, characters of individual frass pellets, oviposition site, mine margins, interaction with leaf architecture, terminal chamber, and exit hole. These characters allow for species-equivalent comparisons of fossil leaf mines, even if the culprits are not known. Leaf mines similar to those on the *Agathis* fossils are otherwise unknown in the fossil record of insect damage that we collectively have now studied through geologic time on several continents from well over about 300,000 specimens. Consequently, the simplest explanation for the presence of similar mines on the same host plant genus at four localities across ca. 18 million years, and not ever on other genera, is the persistence of related leaf miners. However, all of this is only to support the stated idea of persistence as more likely; we still acknowledge the important possibility of convergence as discussed above.

7. “Line 256 ff: Discussion: Here you essentially say that niche conservatism accounts for the long-term associations that originated in the late Cretaceous. I am fine with that but I would like a more in depth discussion of why you think there is sufficient evidence to rule out convergent evolution in these similar types of herbivory, especially in conifer leaves with a high degree of convergent evolution of leaf architecture.”

We agree that the possibility of convergence is needed to be explored further in the manuscript. Reiterating points made above, the overall patterns we observed in this study may have been a result of host tracking, convergence, or both. In our original submission, we attempted to convey this possibility by highlighting both convergence (ecological guilds) and persistence (host tracking). We rewrote and added to the discussion of these concepts (lines 25-29, 77-78, 107-109, 237-261, 341-343, 353-382, 388-390) to increase thoroughness and clarity. We also added to the paragraph on the possibility of convergence (lines 353-382). Our results suggest that ecological guilds have persisted on *Agathis* since the latest Cretaceous or early Paleocene, and we are not claiming that all taxonomic lineages of insects associated with the fossils are still associated with *Agathis* today, allowing for extinction. We do state that host tracking, possibly as a result of niche conservatism, is the most parsimonious and likely explanation for the repeated occurrences of damage types through time, because the entire suite of interactions (not just the blotch mines) is associated with a single, distinctive, persistent host plant genus, *Agathis* and no

other living genus. Convergent evolution cannot be completely ruled out, and could also be a result of environmental stability as a result of host tracking by *Agathis*.

8. “Line 533: Figure 1: I realized that in the guide to insect damage types from 2007, the very same leaf (your Fig 1, D) was used to illustrate DT88. This made me curious about how common this damage type is and whether it is restricted to particular plant groups/plant functional types.”

Damage Type 88 mines have only been found on the *Agathis* fossils described in this paper, which is part of why their persistence on a single fossil host plant genus for ca. 18 million years, plus the presence of analogous mines on extant *Agathis*, is remarkable. As this comment mentions, the blotch mine on *Agathis zamunerae* (Fig. 1D) was originally figured in the 2007 *Insect Damage Guide* (a work that we note was published directly on the Internet as a work-in-progress field guide and not through peer review; a full-length, formal book version is in preparation led by author CCL), to illustrate DT88. The thumbnail description of this damage type in the *Guide* was based on this specimen and others on fossil *Agathis* from Patagonia for field-guide purposes, but detailed work on these mines for formal publication was not undertaken until this project and the present manuscript.

9. “A simple survey in the electronic herbarium catalogue of Edinburgh (herbarium E) shows that pretty similar blotch DTs are occurring on podocarps.

E.g. https://data.rbge.org.uk/search/herbarium/?specimen_num=434594&cfg=zoom.cfg&filename=E00399714.zip”

Thank you for bringing this specimen to our attention. In response, we surveyed the online herbarium of the Royal Botanic Garden Edinburgh for leaf mines on the podocarps *Podocarpus*, *Afrocarpus*, *Nageia*, and *Sundacarpus*. We also looked for leaf mines on *Araucaria*. We had previously surveyed *Nageia* specimens in-person for this project at the herbaria at Royal Botanic Garden Edinburgh and Royal Botanic Garden Kew. We found putative blotch mines on *Podocarpus ingensis* from Bolivia, *Podocarpus oleifolius* from Colombia, and *Podocarpus urbanii* from Jamaica (lines 257-260) with similar morphologies to those on fossil and extant *Agathis* that are now discussed in the paper, adding complexity and interest to the paper as a result of your comment.

10. “Are there distinct DTs on closely related *Agathis*, *Podocarpus*, *Araucaria*?”

We compiled the small number of studies of leaf mining on *Podocarpus* and *Araucaria* (lines 237-257) and also checked the Royal Botanic Garden Edinburgh herbarium catalog for undocumented blotch mines on *Podocarpus* and *Araucaria* (lines 257-260). Both genera have distinct leaf mines. On *Araucaria araucana* leaves in Argentina and Chile, *Araucarivora gentilii* (Elachistidae) mines begin with a serpentine trail that then expands into a rounded blotch mine. We did not find any new blotch mines on *Araucaria*, but did find putative blotch mines on *Podocarpus ingensis* from Bolivia, *Podocarpus oleifolius* from Colombia, and *Podocarpus urbanii* from Jamaica, which are outside of the range of known *Podocarpus* miners (Australasia and Asia). *Podocarpus totara* from New Zealand hosts a curculionid leaf miner (*Peristoreus flavitarsis*), which makes similar mines to those on fossil and

modern *Agathis*. These findings are now summarized and discussed in the manuscript. As discussed in earlier replies, we expanded the section on convergence in the Discussion (lines 353-382).

11. “According to molecular dating studies, this group dates back to at least the Triassic/Jurassic. See e.g. Ran et al 2018 (Proc. R. Soc. B 285: 20181012) Leslie et al 2018 (Am J Bot 105(9): 1531–1544) estimated a clade age for Podocarpaceae + *Araucaria* and agathoids dating back to the early Triassic. Therefore, I would expect a long period of shared evolutionary history. It would be lovely if you could elaborate on this in the light of your findings and explaining to a broader audience if and how insect damage types are diagnostic for different lineages in this clade. This might be useful when discussing the possible long-lasting association between a particular insect (even though unknown) and *Agathis*.”

We agree that exploring the shared evolutionary history of this clade and its insect associations is an important topic to discuss. Currently, there are no known leaf-mine associations on well-defined fossil podocarps or *Araucaria*, so comparisons of the deep time record of their associations are not possible at this time (the Mesozoic conifer blotch-mines we discussed in the manuscript are on other, or enigmatic, lineages that happen to have similar leaf morphology to *Agathis*, and those mines are different from *Agathis* blotch mines as noted above). As just noted, we added a paragraph on leaf mines on related extant conifers, both previously published and found through herbarium surveys (lines 237-261-246). We also expanded on the possibility of convergence of leaf mine morphologies and overall component communities in the discussion section (lines 353-382), as also discussed above.

Reviewer #3 (Remarks to the Author):

“This is an interesting paper and the approach is novel. The have compiled an impressive amount of relevant data as well. The paper is well written and the illustrations and figures are Good and all necessary especially Table 1. I found little editorial problems with the paper. It appears the authors have taken the data at face value and the authors do not wander too far from that data. This is a clean, tight manuscript and recommend publication. When this is published I will make required reading in my evolution and paleobotany class as an example of what kind of biological research is possible on fossils. The paper was enjoyable to read. Recommend publication with little or no changes.”

We thank Reviewer #3 for reviewing the paper and for the compliments.

REVIEWERS' COMMENTS:

Reviewer #1 (Remarks to the Author):

Dear Dr Donovan

Your paper "Persistent biotic interactions of a Gondwanan conifer from Cretaceous Patagonia to modern Malesia" will be very important for the paleobotany and plant-insect interactions field of study, and I think is ready for publication.

Thank you for your "rebuttal letter". As I told you before, I really liked the idea to track plant-insect and fungal associations in *Agathis* through time, and now your results are consistent with the goal and conclusions. I personally would use a different approach, but you achieved the purpose of your research.

Your figures and tables are clear and well organized.

I have no suggestions this time.

Sincerely yours,

Esther Pinheiro

Reviewer #2 (Remarks to the Author):

The revised manuscript has improved a lot and I recommend it to be accepted for publication.

I suggest rewriting one sentence. Page 2, line 35 in the word file:

instead of "and compare it with extant *Agathis*" I suggest writing

"and compared it with damage on extant *Agathis*".

Response to reviewers

COMMSBIO-20-1170-T

Persistent biotic interactions of a Gondwanan conifer from Cretaceous Patagonia to modern Malesia
Communications Biology

Reviewer #1 (Remarks to the Author):

Dear Dr Donovan

Your paper “Persistent biotic interactions of a Gondwanan conifer from Cretaceous Patagonia to modern Malesia” will be very important for the paleobotany and plant-insect interactions field of study, and I think is ready for publication.

Thank you for your “rebuttal letter”. As I told you before, I really liked the idea to track plant-insect and fungal associations in Agathis trough time, and now your results are consistent with the goal and conclusions. I personally would use a different approach, but you achieved the purpose of your research.

Your figures and tables are clear and well organized.

I have no suggestions this time.

Sincerely yours,

Esther Pinheiro

We thank Dr. Pinheiro for reviewing the paper and for providing constructive comments, which helped us significantly improve the manuscript.

Reviewer #2 (Remarks to the Author):

The revised manuscript has improved a lot and I recommend it to be accepted for publication.

I suggest rewriting one sentence. Page 2, line 35 in the word file:

instead of "and compare it with extant Agathis" I suggest writing

"and compared it with damage on extant Agathis".

We thank Reviewer #2 for reviewing the paper and for the insightful comments, which helped us greatly improve the paper. We updated the sentence in Line 31-32 as suggested.